# A Set of Rules for Function-Oriented Automatic Multi-Sentence Analysis in Patents

Christian Spreafico *[ID] and Matteo Spreafico [ID]

University of Bergamo, via Marconi 5, 24044 Dalmine, Italy; matteo.spreafico@unibg.it
* Correspondence: christian.spreafico@unibg.it

**Abstract:** This study proposes some rules for performing a function-oriented search (providing function and object) to extract technical systems from patents, using syntax and dependency patterns to analyse multiple sentences. Unlike the most common inter-sentence analysis methods, the proposed method does not use context information or distance to link the elements of several sentences, but generic terms from patent ontology. The content provided by the rules was entirely derived from a statistical analysis of many patents from different domains, in order to provide a general validity for the rules. The application of the method in two case studies, related to metal cutting and manure processing, highlighted its main advantages. Its degree of automation is such that the expert is almost exclusively excluded, except in the definition of the function on which to build the document pool. The precision and the recall of the results during the tests exceeded 90%. The current limitation concerns the manual control of some results, about 25%, which derive from an additional set of dependency patterns that are difficult to automate and deserve further investigation. The technical systems are many more in number and are more detailed with regard to structural aspects than those obtainable by analysing only single sentences and/or syntax.

**Keywords:** design element extraction; natural language search; inter-sentences analysis; function-oriented search; patent analysis

## 1. Introduction

The study on how to extract knowledge from patents has been very much followed in recent years. This is because the patents contain information for identifying and characterising design solutions, at different levels of detail: functions, behaviours, physical effects, processed objects, devices, materials, structures, forms, etc. [1]. For this reason, many authors studied the automatic extraction of such information from patents, given the enormous amount of data with which to cope. The proposed contributions can be classified in many ways, depending on the logic by which they search for information (e.g., by function or by structure). Other classification criteria can be the degree of automation, the text analysis mechanisms (e.g., syntactic or semantic), the amount of analysed data and the types of extracted outputs. Nevertheless, these methods struggle for affirmation in everyday industrial practice.

This paper proposes a rule-based method for the automatic extraction of technical systems, i.e., devices that perform a given function on a given object, both assigned by the user, from patents. In contrast to traditional syntactic approaches (e.g., [2]), the proposed method exploits a hybrid approach in which dependency patterns are also used. Some dependency patterns have been retrieved in our studies in other fields (e.g., [3]), and retested for the purpose of this method, and new patterns were introduced. Unlike the approaches based on the analysis of a single sentence (e.g., [4]), the proposed method allows us to identify relationships between technical system, function and object in different sentences. Nevertheless, the proposed method cannot be considered an inter-sentence approach, since it is not based on the distance between sentences (e.g., [5]) nor on the

exploitation of specific domain terms provided by the experts (e.g., [6]). In the proposed method, the different sentences of a patent are linked on the basis of common elements, which are generic, patent ontological terms (e.g., method, mean, apparatus), or the same identified technical systems. Consequently, the user has only to define function and object.

The results derived from case studies are also useful for providing some evidence to scientific research fields concerning the topic. The duality in sentence analysis (i.e., syntactic vs. dependency pattern) is useful for comparing the efficacy of the two mechanisms. The dual mode of sentence analysis, i.e., single and multiple sentences, makes it possible to compare the number of results which can be collected in either case, adding experimental material to the strand on intra- and inter-sentence analysis. Finally, the contribution in the function-based search field concerns the investigation of the syntactic lexical links between technical system, function and object, and the level of detail with which the retrieved results are presented. The proposed method can also be useful for the designers looking for technical systems to perform a certain function on an object. In addition, there is no limitation on what the object can be: raw material, semi-finished product to be processed, or finished product to be disposed or recycled. The method can suggest ideas to be developed iteratively whenever a given design problem is reformulated in functional terms, by guiding the design activity.

The objective of the method is to improve the quality and quantity of the results obtained automatically, thus reducing the experts' intervention. The method involved known techniques, appropriately adapted to the field of investigation, and new integrations. Each proposed rule derives from an experimental analysis that was performed semi-automatically on a large patent pool.

The research hypothesis to verify through the case study is whether the values of precision, i.e., the percentage of relevant results out of all those found, and recall, i.e., the percentage of results found out of all possible results, are acceptable compared to other studies. In addition, the usefulness of the integration of dependency patterns for syntactic analysis and the use of multiple sentences have also been investigated.

The paper is organized as follows. Section 2 presents the literature review about the previous approaches working in this field. Section 3 presents the proposed method, including an introduction about the searched design elements, the main points on which it is structured, and an explanation of the most appropriate way to use it. Section 4 shows how the method was applied in two case studies and presents and discusses the results. Finally, Section 5 draws the conclusions.

## 2. Literature Background

In the last few years, many methods to automatically analyse documentary sources (e.g., scientific articles and patents) to extract design elements (e.g., functions, behaviours and structures) have been developed. These methods are mainly based on natural language processing (NLP), to analyse unstructured texts and to classify the extracted information. NLP is generally preferred to other approaches, due to its documented efficiency in extracting the design elements from different sources (e.g., [7,8]). In particular, in more recent contributions, NLP has also been applied to analyse surveys, experts' interviews, quantitative data and consumer opinion data (e.g., [9]), while to reorganize the extracting information, knowledge management techniques have been integrated (e.g., [10]). The main limitation of all these approaches is the involvement of an expert to discuss, confirm and contextualize the obtained results. Only in this way, can the pertinence and accuracy of the obtained results reach appreciable levels of up to 90% (e.g., [11]).

NLP approaches are generally based on keyword searches and the grammatical or syntactical analysis of the sentence (e.g., [12]). Their working principle is the association of syntactic roles (e.g., subject, verb, object) with the words of the sentence, using syntactic parsers, and ontological roles (e.g., application field, working principles, materials), using syntactic roles (e.g., [2]). More in detail, through syntactic parsing methodology, a sentence is simplified, by identifying the minimum number of syntactic elements and hierarchizing

them, e.g., through the subject–action–object (SAO) triad. In this way, complex sentences became more comprehensible, especially for designers and technicians [2]. However, SAO triads cannot be used to extract all kinds of technical features. For this reason, specific search strategies can be integrated into their identification [13]. Among these, the parsing procedure is based on the computation of the relations among the words that compose a sentence (called tokens), and encodes a graph of their syntactic relations [4]. Such relations are limited in number, and dependent on the software adopted for the analysis. The main open problem of these approaches deals with the management of polysemy, i.e., the words having different meanings, and synonymy.

Other NLP-based approaches exploit artificial intelligence. Among these, some expert systems emulate human reasoning to identify hidden patterns within the sentences and extract data from them (e.g., [14,15]). Their validity has been testified especially in analysing unstructured documentary sources, where traditional linguistic patterns are not included (e.g., [16,17]), while their main limitation concerns the classification of grammatical elements ([11,18,19]).

Another common limitation of all the described approaches is their ability to analyse only a single sentence to extract relations among elements. As a consequence, the extracted relations are fewer, since many of them involve elements of multiple sentences. In this regard, the analysis in [20] showed that 28.5% of the relations in a text occur between entities from different sentences. The capturing of the relations between elements from different sentences is more difficult, since complex semantic relationships are involved and complex syntactic and semantic dependencies are required. This is because existing methods retrieve only some relationships, exploiting only some dependencies ([6,21]). According to [5], the reason for this problem is the absence of a comprehensive study and a large dataset including the dependency patterns of elements from different sentences. However, all previous studies in the literature explored them only in specific application fields. As result, databases with very limited content and which are too specific to be compared have been proposed.

Some attempts to overcome these limitations are the mixed method approach (e.g., the kernel approach and NLP with POS tagging), implementing dependency patters to analyse single and multiple sentences. However, the obtained results do not significantly exceed in number those obtained by the approaches analysing a single sentence ([6,21]). Other approaches combine the dependency patters with specific terms extracted from the descriptions of the Cooperative Patent Classification classes or other sources of technical knowledge (e.g., [22–24]). However, their main limitations are the narrowness of application areas and the need for continuous updates.

## 3. Methods

The proposed method allows for the automatic identification, within a patent pool, of devices performing a certain action on a certain entity. These design elements (i.e., the device exercising the action, the action itself and the entity undergoing the action) can be reported in a multitude of ways in patents. For this reason, the ontology of the TRIZ (the Russian acronym for the theory of inventive problem solving) minimal technical system (MTS) model has been implemented within the method [25].

The TRIZ-MTS model was chosen because it provides all the elements to categorize information extracted from patents in a practical way in order to support problem-solving/design. There are some alternative models already used to categorize information extracted from patents, also automatically, and combined with TRIZ, such as Function-Behaviour-Structure theory [26] or failure mode and effect analysis (FMEA) [27]. However, such models work at too high a level of detail. While they classify information in relation to both function and structure, they fail to go down to the level of the components of the structure and their connection to function, as in MTS. Such aspects, on the other hand, are essential to adequately support the designer in intervening on the structure, by exploiting the information extracted from the patents.

The action is more properly defined as a unction, which is a transformation performed on the entity, defined as an object, to modify it into a so-called product. The definition of the device that performs the function is broader and more comprehensive, not just limited to its structure, but using the term technical system. The latter consists of four main parts. The tool is the part that is in direct contact with the object during the performance of the function, according to a mechanical, acoustical, thermal, chemical, electrical, magnetic, intermolecular, or biological interaction. Supply is the part that generates the energy for performing the function, the transmission is the part that transmits energy from the supply to the tool and finally to the control, which regulates the operation of the technical system by interacting with the other parts.

The definition of the technical system is sufficiently broad to model devices that are significantly different from each other. For example, in the case of a hammer, the technical system includes both the device and the user, the tool is the head of the hammer, and the transmission is the set of the hammer handle, hand and arm of the user. The supply is the user muscle, and the control is the sight and hand of the user when he directs the hammer head exactly onto the nail (object) to hit it (function). In the case of a computer numerical control (CNC) machine, the technical system coincides with the device, since the tool is the utensil, the transmission is the axis and kinematics that move it, the supply is the set of electric motors that rotate and move the utensil, and the control is the set of the automatic sensors.

Figure 1 provides a schematic representation of the minimal technical system model.

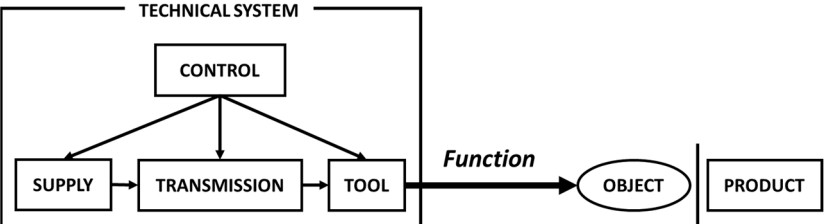

**Figure 1.** Minimal technical system model [25].

Table 1 reports the definitions of the considered main ontological elements. More explanations for each element are provided in Table A1 in the Appendix A.

**Table 1.** Ontological elements used to define the methodological steps [25].

| Ontological Elements | Definitions |
| --- | --- |
| Technical system | The machine (e.g., a device or a plant) searched by the designer to solve the design problem. |
| Function | The action performed by the technical system. |
| Object | The entity over which the technical system performs the function. |
| Product | The transformed object after the function has been performed. |
| Tool | The part of the technical system that is in direct contact with the object during the performance of the function. The contact can be mechanical, acoustical, thermal, chemical, electrical, magnetic, intermolecular, or biological. |

The objective of this study can therefore be reformulated in line with this ontology, as the automatic determination of the technical systems from patents to the performance of a function on an object assigned by the user.

To fulfil this objective, the proposed method consists of two main steps, presented in detail in the following. The first one consists of the determination of the document pool to be analysed, which is strictly dependent on the definition of the problem space, that is, how the initial problem to be solved is defined. The second step consists of extracting the design elements from the pool of documents.

### 3.1. Problem Space and Pool Definition

In design, the definition of the starting problem is a fundamental task, since it can greatly influence the search for solutions. This is because in one of its most established definitions, design is considered merely a problem-solving activity, where the starting problem coincides with a problem space. This latter contains all the details that define the problem, and the constraints and boundary conditions, while the alternative solutions are contained in the solution space [28]. In this regard, the extent of the solution space depends on the extent of the problem space, as well as on the design methods, knowledge, and creativity of the designer.

In the proposed approach, the design knowledge extracted from patents is strictly dependent on the definition of the problem space regarding the definition of the search query to be used in the patent search. Having established the correctness of the formulation of the starting problem, which is not the task of this study to discuss, the first tedious task that arises in the application of the proposed method therefore concerns the definition of the search query.

One of the best ways to define a search query is to include a verb and a direct object. They are, respectively, the function and the object described in Table 1. This strategy is in line with the correct definition of the design problem according to the TRIZ method, because in this way any reference to tool and technical system is avoided, and they are instead the design goal, i.e., the solution, and any inertia that may affect the design activity. The same reasoning is also valid at the level of patent search, allowing for the enlargement of the recall of the identified documents and therefore the solution space [24]. In many cases, however, the problem may be far from trivial, given the many possible ways of understanding the same function in relation to the level of detail and the repercussions that its definition may have on the retrieved results obtained [29].

Knowledge of the problem space is therefore also fundamental in order to allow the expansion of the query, including in it synonyms of both the verb used for the function and the noun used as object. This option makes it possible to greatly expand the results obtained in a database search, and the possible solutions, but only if the synonyms are carefully selected. There are several supporting tools, such as the Oxford Dictionary, for expanding query terms by looking for synonyms and modifiers.

Finally, the effectiveness of such a query also depends on the possibility of syntactically relating function and object as verbal predicate and direct object. Many document databases allow the query to be defined in this way, while others guarantee the possibility of establishing a semantic relation, for instance by establishing or including them in the same sentence with a distance operator. To retrieve design solutions, patent databases are suggested. This is because patent databases contain most of the world's patents, i.e., over 110 million. In addition, the search can be streamlined when conducted in subsets, such as the WIPO or USPTO, or the entire EP, which counts a smaller number of patents.

### 3.2. Technical Systems Extraction

The objective of this step is to extract the technical systems from the collected patents with the query formulated in Section 3.1.

The starting hypothesis of this study is that all the design elements considered, both those that are to be obtained (i.e., the technical system) and those used to obtain them (i.e., function and, object) have precise syntactic roles within the sentences of the analysed documents. Furthermore, it is also assumed that there are a finite number of lexical forms (e.g., by, in order to) and nouns (e.g., method, mean, system), which are used recursively to link design elements at the grammatical level.

Based on this assumption, the proposed method uses a semantic and syntactic analysis of patents to determine the technical systems, performed automatically using NLP (natural language processing) tools. To use these tools correctly for this analysis, however, it is necessary to define a rule, the purpose of which is to explain how the technical system can be related to the elements of the sentence. For instance, in the phrase: "A laser cuts a

metal by evaporation", "laser", which is the technical system, is also the subject, "cut" is the function and the verbal predicate of "laser", and "metal" is the object and the direct object. If the sentences were all formulated in this way, then only one rule could be defined for our analysis. However, not all design elements always appear in the same sentence, but may also be scattered over several sentences and still be logically connected. Consider, for example, the following two hypothetical sentences from the same patent text: "The claimed device cuts a metal" and "Said device is a laser".

For this reason, a few rules have been developed in this study to recover more technical systems, even in cases such as the second example.

Each rule provides three types of directions to be followed:

1.  The identification of which sentences the rules work on to extract technical systems, depending on the other design elements contained therein, i.e., other technical systems, functions, objects, and other lexical elements.
2.  What syntactic roles the design and the other lexical elements can have in the various sentences, with the aim of isolating the possible technical systems with automatic logical analysis.
3.  Which dependency patterns can be used in sentences to express design and other lexical elements to automatically refine the results of the logical analysis, improving the precision of the results.
4.  In the following paragraphs, the three rules are explained in detail.

### 3.2.1. Rule 1

Rule 1 explains how to extract the technical system from a single sentence (called the main sentence) in which the function and object also appear (see Table 2). This choice is one of the most common, which is typically considered by most automatic text analysis methods.

**Table 2.** Content of the sentences considered by Rule 2.

| Used Main Sentence | Example |
| --- | --- |
| Function AND Object AND Technical system | The laser (Technical system) cuts (Function) a metal (Object) |

At the syntactic level, the hypothesis on which Rule 1 is based is that, in a sentence, the technical system is the subject, the function is the verbal predicate, the object is the direct object, and the behaviour is an indirect expansion sustained by certain lexical forms. With this logic, the extraction of the technical system and behaviour can then be performed automatically by tools that are able to perform logical sentence analysis.

Most NLP tools that can process a sentence like the one considered by Rule 1, require a lemma (i.e., noun, verb, adjective) as input from the user, and use it to search for terms related to it syntactically or semantically. Concerning the choice of the lemma, the function-oriented approach [24] suggests using directly the verb expressing the function for the greater effectiveness in determining the SAO (Subject–Action–Object) triads to be considered according to Rule 1, i.e., those in which the subject is the technical system and the direct object is the object.

This is because not all SAO triads containing the function and object also contain the technical system. The cases where the subject is not the correct one are different, according to the research carried out for this study.

The technical system may not be recognised correctly when expressed by two terms: e.g., in the sentence "The cutting wheel cuts a metal layer", an NLP tool based only on syntactic analysis can only return as a subject "cutting", but not "wheel", effectively losing the information that is most needed to define the technical system, or, when two technical systems execute the function, e.g., "A mold and a counter-mold cut the metal", only one of the two could be automatically recognised as a subject.

Finally, there are all the sentences in which the subject is not a technical system, e.g., "the table (subject) for cutting metal includes a circular saw (technical system)".

All these errors are strictly dependent on the software being used. For instance, with Sketch Engine (https://www.sketchengine.eu/) (accessed on 23 July 2023) software all those reported have been found experimentally, while spaCy (the open-source library for NLP in Python) is able to identify subjects defined in a much more complex way, including structural characteristics. For instance, in the sentence "the wear-resistant silicide-based cermet tool to cut a metal is made from the following raw materials in parts by weight: 50 parts of titanium silicide, 15 parts of zirconium silicide", spaCy identifies, as subject, "the wear-resistant silicide-based cermet tool is", rightly attaching to it also "50 parts of titanium silicide, 15 parts of zirconium silicide" as an appositional modifier.

In any case, to facilitate the identification of the technical systems within Rule 1, our proposal is therefore to also perform a semantic processing of the sentences obtained with the syntactic analysis, by launching an automatic sub-routine. The latter exploits certain dependency patterns (see Table 3) that typically introduce a technical system. They consist of the union of the function and some lexical forms.

**Table 3.** Where TS = Technical system.

| Dependency Pattern | Example Sentences |
|---|---|
| TS + for + Function | The rotating blade for cutting metal components (TWI645528B). Milling cutters for cutting metal billets (RU2678554C2). |
| TS + to + Function | Plasma arc torch to cut metal (US9789561B2). A laser to cut metal tubes (US20200170793A1). |
| TS + used for + Function | Cutting torch used to cut metal objects (US9836994B2). A laser beam is used to cut metal (US9576932B2). |
| TS + used to + Function | Continuous wave laser used to cut metal (US8800475B2). $CO_2$ laser used to cut metal (US8800475B2) |
| Function + by + TS | The metal plate is cut by the gas torch (JP6182666B2). Metal cut by saw (US9576932B2). |
| Function + with + TS | Metal cutting with the rotary blade (JP6356610B2). Metal cut with a laser (US9107725B2). |

The starting point for the collection of these dependency patterns was the analysis of the patterns collected for other purposes, i.e., to automatically catalogue risk assessment methods from scientific papers [30] and to extract functions and application fields from patents [3]. In this work, a new analysis was carried out ad hoc to confirm the validity of the dependency patterns retrieved from the literature, through the following procedure.

First, a patent pool related to metal cutting was analysed. The used query was "((cut+ or divid+ or slit+ or chop+ or separat+ or lanc+ or hash+ or sever+ or cleav+ or rend+ or sunder+ or dissever+) s (metal+))/TI/AB/TX", which was launched in the Fampat patent database. Just over three million patents were retrieved through this query. Patents with priority dates in 2016 were arbitrarily excluded from this pool. To simplify, only the queries of these patents whose sentences correspond to about 20% of the total sentences in the pool were processed using an NLP tool.

A search on the web identified 314 technical systems for metal cutting. This set includes both generic technical systems, e.g., lathe and milling machine, and more specific ones, e.g., $CO_2$ laser. All 314 technical systems were then used as input for the automatic syntactic processing of the documents. A total of 185 were identified in the analysed pool.

All the sentences in which these 185 Technical systems, syntactically linked to the function "cut" and the object "metal", were manually analysed. From all these sentences, only those containing dependency patterns linking the technical system to the function have been isolated. In this case, the resulting technical systems were 127. The remaining sentences contain SAO triads about the technical system, function and object, without dependency patterns, e.g., "A laser cuts metal".

Finally, all the identified dependency patters, i.e., those used in conjunction with the 127 technical systems, were collected.

Table 4 shows all the analysed dependency patterns, from which the most common ones were extracted and reported in Table 3.

**Table 4.** Analysed dependency patterns (where TS = Technical system).

| Dependency Patterns | | Identified Technical Systems with the Dependency Pattern | Percentage Compared to Total Number of Identified Technical Systems (185) |
|---|---|---|---|
| Known relations (from [3]) | TS + to + Function | 33 | 18% |
| | TS + Used for + Function | 27 | 15% |
| | TS + For + Function | 19 | 10% |
| | TS + Used to + Function | 19 | 10% |
| | TS + Used of + Function | 10 | 5% |
| | Function + like/as/of + TS (e.g., A device for cutting like a lathe) | 4 | 2% |
| | TS + Can + Function | 4 | 2% |
| | Total | 116 | 62% |
| New relations | Function + by + TS | 21 | 11% |
| | Function + with + TS | 15 | 8% |
| | Total | 36 | 19% |

From this analysis, some alternative dependency patterns were collected (see Table 5).

**Table 5.** Dependency pattern of the Object of the main sentence.

| Dependency Pattern | Examples |
|---|---|
| Object (noun) + noun | Metal workpiece/piece, sheet metal |
| Object (adjective) + noun | Metallic sheet |
| Noun + made of/consisting of/realized by + Object | Workpiece made of metal |

### 3.2.2. Rule 2

Rule 2 explains how to extract a technical system from an incomplete sentence in which it appears without the function and/or object, using another auxiliary sentence in which these two elements appear. The auxiliary sentence generally appears before the sentence used, within the patent text. According to this rule, the mode used to link the technical system of the used sentence to the function and object of the auxiliary sentence is the presence of a generic technical system, i.e., a generic term (e.g., "method", "mean", "system") commonly used in patents to indicate a technical system. A sub-variation of Rule 2 concerns the extraction of a new technical system from the incomplete sentence, using an auxiliary sentence specifying which technical system is referred to in the generic technical system.

Table 6 summarises the criteria for selecting the sentences considered in Rule 2.

**Table 6.** Content of the sentences considered by Rule 2.

| Used Main Sentence | Used Auxiliary Sentence |
|---|---|
| Generic Technical system (GTS) AND New Technical system (TS)/Technical system <br>• Said machine (GTS) can also be laser (NTS) <br>• Said method (GTS) can also be a milling machine (TS) | GTS (AND Technical system) AND Function (F) AND Object (O) <br>• A lathe (TS), called machine (GTS), to cut (F) a metal (O) <br>• A method (GTS) to cut (F) a metal (O) |

At the syntactic level, in the auxiliary sentence, the generic technical system is the subject, the function is the verb, and the object is the direct object, while in the main sentence there are two possibilities: the generic technical system is the subject and the technical system to be retrieved is the direct object, or vice versa.

Rule 2 was developed because it is based on a recurring stylistic pattern in patent writing, constructed from the use of few sentences in which the technical system, function and object are stated, and which provides specifications for other sentences. The aim is usually to broaden the scope of protection that a patent can offer at the legal level by claiming as many design variants as possible. This mechanism is very common, especially in the claims, although it is not missing from the description. An independent claim defines the essential characteristics of the invention whose protection is required, and serves to identify the invention. A dependent claim contains all the features of the independent claims to which it is linked, and indicates further features or variants for which protection is sought. It is not uncommon for a patent to have several independent claims and multiple dependent claims linked to them. Such alternative technical systems can be very useful during design, as they include real alternatives (e.g., lathe vs. milling machine), models and variations of the same device (e.g., numerically controlled lathe), and characteristics of its parts, such as materials and geometries (e.g., conical tip).

For our purposes, the analysis of all these "secondary" sentences (here called main sentences), whether they are part of the description or dependent claims, is very useful, as many alternative technical systems are included, often even in a single sentence. However, in these sentences, alternative technical systems do not always appear together with the main technical system, i.e., the one that is syntactically and/or semantically linked to the function and object in another sentence. Many times, patent writers prefer to use generic substitutes for both the technical system and the object, instead. The former is used by Rule 2 to identify technical systems.

The following procedure was used to identify these generic substitutes, which have been renamed as the "generic technical system".

In the same patent pool used to determine the dependency patterns of Rule 1, only patents containing a limited set of technical systems, defined randomly, were automatically isolated, to limit the analysis. They are CNC, drill, laser, oxyfuel, plasma, and torch.

All the sentences of each patent that contain one of the used technical systems (i.e., the main sentences in Table 2) and those containing the function, expressed with the root "cut" (e.g., cut, cuts, cutting,) and the object described by the root ("metal" (e.g., metal, metals, metallic plate) (i.e., the auxiliary sentences in Table 6) were automatically collected.

These sentences were then processed individually in spaCy software [4], using it as a dependency syntactic parser. The collected results were then manually analysed.

A generic technical system has been collected, for when three conditions exist simultaneously, as established by Rule 2:

1. In the same patent, at least two sentences have been identified, one of the main-sentence type and one of the auxiliary-sentence type.
2. From spaCy's analysis of the main sentence, the generic technical system is the subject. The verbal predicate is a verb, which may be combined with a syntactic particle, such as "is, can be, consist of, is made of, comprises, etc.". The direct object is one of the considered technical systems.
3. From spaCy's analysis of the auxiliary sentence, the same generic technical system is the subject, the function "cut" is the verb, and the object "metal" is the direct object.

Consider, for example, the two sentences taken from the patent CN107009098A, in which the technical system is "CNC milling machine", which was used to extract the generic technical system "method". The main sentence is "The method according to claim 1, wherein the processing tool comprises a CNC milling machine and performing the wire cutting process". The used auxiliary sentence is the claim 1: "A method for cutting sheet metals". Both the results satisfy the requirements of point 1. As can be seen from Figure 2, taken from the main sentence analysis carried out with spaCy, the term "method" was

recognised as the subject, "comprises" as its verbal predicate and "CNC milling machine" as its object, in line with point 2. Meanwhile, from Figure 2, taken from the analysis of spaCy's auxiliary sentence, it can be seen that "method" has been recognised as the subject, "cutting" as the verbal predicate, and "sheet metal" as the object, as established by point 3.

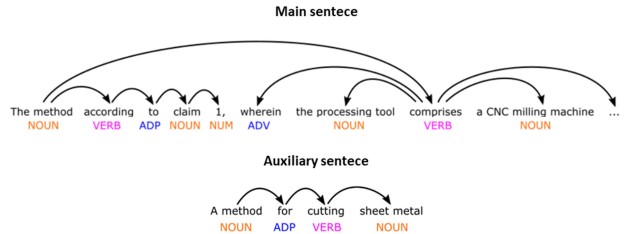

**Figure 2.** Examples of results of automatic processing of two sentences.

Following the same procedure for all the collected sentences, the generic technical systems shown in Table 7 were identified. They are generic nouns (e.g., method) that can be preceded by an introductory particle (e.g., said).

**Table 7.** List of Generic Technical Systems.

| Introductory Particle | Said, the, the Described, This |
|---|---|
| Generic noun | Apparatus, artefact, body of invention, component, device, mean, mechanism, member, method, object, object of invention, part, process, product, system, what is claimed. |

At the same time, from the automatic analysis of the main sentences (point 2), it was also possible to isolate the verbal predicates that are most used to link the technical system with the generic technical system (see Table 8). They can be used independently, as the generic technical system is the subject and the technical system is the verb, and vice versa. In general, all these dependency patterns are valid to help automatically extract technical systems, except for those in the last three lines, categorised as "additional". The latter are, in fact, aimed at introducing constitutive elements of the claimed device, which may be able to perform the function, and therefore be definable as a technical system, or not, e.g., "fluid-cooling device". Such elements may, however, be useful for design purposes if they are associated with the cutting device of which they are part, and not considered as its substitutes. With a view to automating the method, this fact should be duly considered to avoid errors when cataloguing the results.

**Table 8.** Dependency pattern of the verb of the main sentence (where * = Truncation e.g., compris*: comprises, comprising).

| Dependency Patterns | Examples |
|---|---|
| is/can be | ... where said method is a 3D/3 axis or 2D/2 axis plasma |
| is + in/off | ... knives are of the conical configuration |
| is generated by | ... wherein said method is generated by Ytterbium laser beam |
| is defined by/in/as | ... wherein said method is defined by CNC control lathe machine |
| is formed/realized + in/by | ... an annular water inlet groove is formed in one side of the device |
| is made + of /up/with | ... wherein said method is made of oxyfuel cutting ... |
| consist* of/in | ... in which said method consists of: plasma cutter, ... |
| Additional | |
| compris*/comprised by includ*/is included in embed*/is embedded in | ... the machine tool comprises a drill, surface grinder, cylindrical grinder, embroidery grinder ... |
| has/hav* | ... wherein said method has: arc torch, a fluid cooling device, ... |
| is provided with | ... part is provided with a circular cutting knife |

From the analysis of the auxiliary sentence (point 3), instead, the dependency patterns of the verbal predicate (function) used to link the generic technical system to the object, were determined (see Tables 8 and 9). A reassurance of the completeness of these dependency patterns is given by the work of [3], who, analysing a larger and more heterogeneous pool of documents, proposed the same list.

**Table 9.** Dependency pattern of the verb of the Auxiliary sentence (where GTS = Generic Technical System, O = Object).

| Dependency Patterns | Examples |
| --- | --- |
| GTS + for + Function + ing + O | Method for cutting metal |
| GTS + to + Function + O | Method to cut metal workpieces |
| GTS + can + Function + O | A laser-beam method that can cut metal |
| GTS + used for + Function + ing + O | A method used for cutting metal pieces |

Finally, the analysis of the dependency patterns of the object in the auxiliary sentence did not reveal any new results compared to those shown in Table 4.

### 3.2.3. Rule 3

Rule 3 explains how to obtain a new technical system from a sentence in which it appears together with another technical system because it has already been identified by analysing other sentences with one of the rules already presented. The auxiliary sentence used in this is the main sentence used in Rule 1. For this reason, the technical system is already known, and through the application of Rule 1 has been defined as the subject of the function searched and executed on the object. The particularity of the main sentence is that in it the new technical system is defined as an alternative to, or a constituent part of, the technical system (e.g., supply, transmission or tool).

Table 10 summarises the criteria for selecting the sentences considered by Rule 3.

**Table 10.** Content of the sentences considered by Rule 3.

| Used Main Sentence | Used Auxiliary Sentence |
| --- | --- |
| Technical system AND New Technical system<br>• The laser (TS) can be a $CO_2$ laser (NTS) | Main sentence of Rule 1 |

The justification for adding Rule 3 is the same as that for Rule 2, i.e., the habit of adding several alternative technical systems to the main one, i.e., the subject matter of the invention of the patent. The way of working is different, however, since Rule 3 concerned those sentences in which the main technical system appears, together with alternative ones.

In order to better understand how these sentences are constructed, i.e., which dependency pattern they use, the same experimental analysis carried out to define Rule 1 was performed, since in that case we had already isolated those which for Rule 1 were the main sentences and for Rule 3 become the auxiliary sentences. In addition to these, the main sentences of Rule 3, i.e., the sentences in which the main technical system is expressed, together with other alternative technical systems, were also collected.

In order to isolate the sentences to be analysed, we simply extracted all the sentences containing the 185 technical systems which appear in the auxiliary sentences (i.e., the main sentences of Rule 1). These technical systems are the main technical systems of the main sentences. Then, by manually analysing all these sentences, only those containing the alternative technical systems were extracted, and were automatically analysed with spaCy, to identify the dependency patterns present.

The result of this analysis is that the dependency patterns of the verbal predicate that are used to link alternative technical systems to the main ones are exactly all those already expressed in Table 8.

### 3.2.4. Rule Application

The discussion of the application of rules must take into account two aspects: how many of them are used, i.e., some or all of them together, and in what order. Regarding the number, we must consider that Rule 1 and Rule 2 are independent, while Rule 3 presupposes the use of Rule 1, using its main sentences as auxiliary sentences. Therefore, the way the rules are constructed means that their order of use is rather constrained, leaving few combinations open, especially if all three are used. Another aspect to consider is the available resources, since each rule requires additional analysis. However, the more rules one considers, the more technical systems can be identified. Assuming that all the rules are used, and by virtue of this reasoning, a possible sequence of use could be the one with which the rules have been presented. From this perspective, considering a patent pool, the application of Rule 1 allows us to collect a first set of technical systems. Through Rule 2 and Rule 3, they can retrieve new ones, which are reported in sentences not completed with all design elements, such as those analysed with Rule 1, thus expanding the set of results obtained.

Figure 3 offers a graphic schematisation of this application of the rules.

The application of the rules and the help they can provide to the design activity depends on the logic with which the search for the technical systems, which they aim to improve, is carried out, i.e., specifying the function and object. In response to them, the rules help to retrieve the technical systems related to them. In this type of search, if one narrows down the field by using a more specific function and/or object, one can obtain more specific technical systems, and vice versa. Another possibility is to set up a search to better explore the characteristics and constituent parts of one of the technical systems that have been identified. For example, instead of searching for technical systems related to the function "cut" and the object "metal", it is possible to construct a pool based on the function "move" and the object "cutting tip", to retrieve which transmission and supply can be used. The iterative application of the proposed method in relation to this multiple search logic is a possibility for fully supporting the design in the retrieval of a knowledge base to search for solutions. Such a way of proceeding could then be best grafted onto a sequential design approach (e.g., [31]), where a problem is broken down into sub-problems and addressed sequentially.

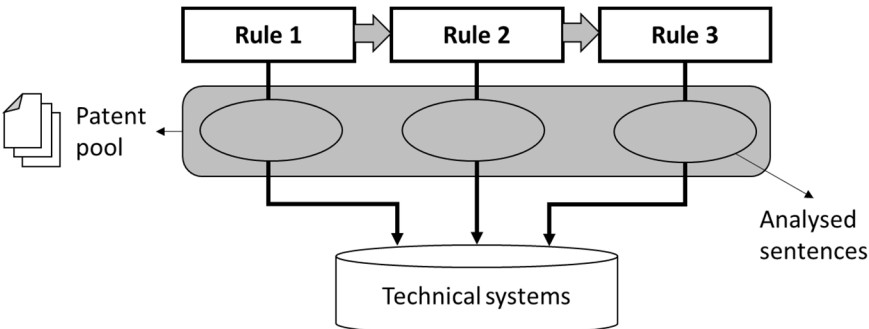

**Figure 3.** Representation of the operating mechanism of the proposed rules.

## 4. Case Study

The objective of this case study is to evaluate the application of the proposed method on a sample of documents, with respect to the following criteria.

The number and the characteristics of the obtained results, i.e., technical system, in relation to the different rules and their suggestions, i.e., usefulness of syntactic analysis and dependency patterns (Section 4.2).

The recall of the method, i.e., how many results the method allowed us to identify, with respect to all those contained in the sample (Section 4.3).

The precision of the method, i.e., how many results can be considered correct.

To obtain these evaluations, the analysis was automatically performed, and the results were manually checked by the authors. This also made it possible to understand to what extent and how the method is automatable at present.

All evaluations were carried out within a case study still related to metal cutting, although based on a different pool of documents than the one used in the proposal. The test was then repeated for another case study in a different field, i.e., manure processing, to compare the results and evaluations (Section 4.4).

### 4.1. Test Execution

The objective of the first case study is to extract the technical systems that perform the function "cut" of the object "metal".

To build the pool of documents to be analysed, only patents with priority dates in the year 2016 were used, which were not previously considered. The query used in the FAMPAT patent database was "((cut+ S metal+) AND eprd = 2016)/TI/AB/TX" in full text. However, to be able to manually check all the results obtained automatically following the rules for reliability, only a selection of 5000 patents out of the total 40,780 were considered, with the most recent priority date.

Therefore, all the patents considered were processed automatically with the software Sketch Engine, which allows us to perform the syntactic analysis of the sentences by means of parsers and semantics, returning a series of outputs (nouns and verbs) catalogued according to syntactic role (i.e., subject, verbal predicate, and object) and pattern. This software was selected to carry out the test since, thanks to the output offered, it is possible to identify technical systems and assess the usefulness of all the criteria established by the rules.

In collecting technical systems, the three rules were applied in sequence, according to the logic presented in Figure 3. Only new technical systems were counted for each rule: those found by Rule 2 that had already been identified by Rule 1 were not counted, nor were those identified by Rule 3 that had already been found by Rule 1 or Rule 2.

In Rules 2 and 3, the link between the main sentence and the auxiliary sentence was performed manually by checking, among the Sketch Engine results, the identification number of the document from which the two sentences were taken.

### 4.2. Overview of the Results

The number of technical systems automatically collected by this analysis and positively evaluated following the manual check was 117. Of these, 32 were identified with Rule 1, a further 45 were added with Rule 2, and a further 40 with Rule 3. This result is in favour of the method, as it highlights the usefulness of Rule 2 and Rule 3 in considerably expanding the set of results (i.e., the technical system), compared to those recovered with Rule 1.

One issue to be discussed concerns the level of detail with which technical systems are presented in patents, which has not been mentioned so far. Without claiming to classify technical systems based on strict theoretical criteria relating to structural and functional aspects, we have simply divided them into two levels, according to the name obtained from the analysis. The first contains technical systems of all types, defined by their generic name, e.g., laser, plasma, cutting wheel. The second level is more specific, since it can refer to a characteristic of the technical system or to a model of it. Such technical systems have been identified by their binomial names, e.g., "pulsed laser" or by a modifier within the main sentences considered by Rule 2 and Rule 3, e.g., "The laser can be the CO2 type".

The map in Figure 4 shows all the technical systems identified and classifies them according to the rule with which they were collected, and their level.

Figure 4 shows that Rule 1 resulted in the highest number of first-level technical systems, i.e., 21, while Rule 2 resulted in the addition of 7 new ones at this level. Among the second-level technical systems, on the other hand, the advantages of applying both Rule 2, which led to the identification of 38, and Rule 3, which led to the identification of 40, can be clearly seen, while Rule 1 led to the identification of only 11.

In this case, Rules 2 and 3 therefore proved to be particularly advantageous in expanding the set of useful results, but above all in offering specifications relating to technical systems, working mainly on the second ones, undoubtedly during the design. By recalling the definitions of the MTS model, it is possible to better understand what this information is. The top-level technical systems are divided into aggregates (i.e., the actual technical systems), e.g., "punching machine, lathe," and e-tool, e.g., "hot cutter, wire, plasma, cutting disk". The second-level ones can express details about the tool, such as the material, e.g., "diamond milling, steel wire", and the shape, e.g., "hook-shaped cutting teeth, V-shaped notch". In other cases, details are given of other parts, such as the supply, e.g., "diode laser". Others specify a way of using the technical system, e.g., "pulsed laser". Finally, the types of technical system can also be specified., e.g., "vertical milling, cermet lathe".

In order to assess the usefulness of the dependency patterns developed for the various rules, the test was to see how many of the total technical systems could be derived using the rules but only with syntactic analysis, i.e., without the dependency patterns. This analysis was carried out automatically, using Sketch Engine, and counting only the technical systems appearing among the subjects and objects or their modifiers and adjectives.

The number of technical systems collected in this way was 25, i.e., almost 22% of the total, and they are reported in Table 11, where they have also been classified according to their level of detail. This result therefore indicates that in all other cases (i.e., 78%) the proposed dependency patterns were used, thus confirming their obvious usefulness.

In particular, as shown in Table 11, 16 technical systems were identified with the syntactic analysis belonging to the first level, while only 9 belonged to the second level. This means, therefore, that dependency patterns, in this test, proved to be useful, above all in determining the second-level technical systems, where the effectiveness of the syntactic analysis proved to be much lower than in the first-level ones, i.e., 10% (9/86) vs. 57% (16/28).

*4.3. Testing the Recall of the Method*

By manually analysing all the documents, it was possible to identify exactly all the technical systems present in the analysed pool, even those reported in the sentences that were not considered or analysed by the proposed method. The percentage ratio between the technical systems identified with the method and the total is the recall from the method itself in this case study, which was equal to 97%. This is because, by means of the manual analysis, 4 technical systems not identified by the method were determined (see Table 12), out of a total of 121.

Analysing the sentences reported in Table 12, some reasons for the exclusion of the technical system were identified. The sentences relating to the first two technical systems were excluded a priori from the analysis, since their syntactic structure is not covered by any rule. In fact, in both cases, the new technical system (i.e., the conical shape and the rake cutting blade) is not linked by any verb, nor by one of the dependency patterns proposed, to the known technical system (i.e., multi-edge cutter and the main cutting blade), as it should be instead in the main sentence of Rule 3. In addition, in these sentences the function, expressed in verbal form, and the object, as in the main sentence of Rule 1 and in the auxiliary sentences of Rules 2 and 3, are missing. However, for these, the syntactic analysis fails to find relations between the new technical system (e.g., cutting edges, diamond wire) which is the subject and the technical system (drill rod, wire-winding drum), which should be the object, as established by Rule 3, since the object is instead another item (i.e., circumferential direction, spiral grooves).

The proposed method extracts the technical systems that perform a given function prescribed by the user. Then, the retrieved technical systems can be used, as input, in other methods to determine the new functions they perform. Finally, using each new function, the proposed method can identify new technical systems. For this purpose, the proposed method can then be integrated with other feature-extraction methods, e.g., those based on SAO triads, previously described in Section 2—Literature Background.

### 4.4. Testing the Precision of the Method

The precision of the method was assessed in two ways. In the first case, the correctness of the automatically extracted technical systems was tested using almost all the suggestions in the rules, except for the additional dependency patterns in Table 8. In the second case, only those extracted with these dependency patterns were tested.

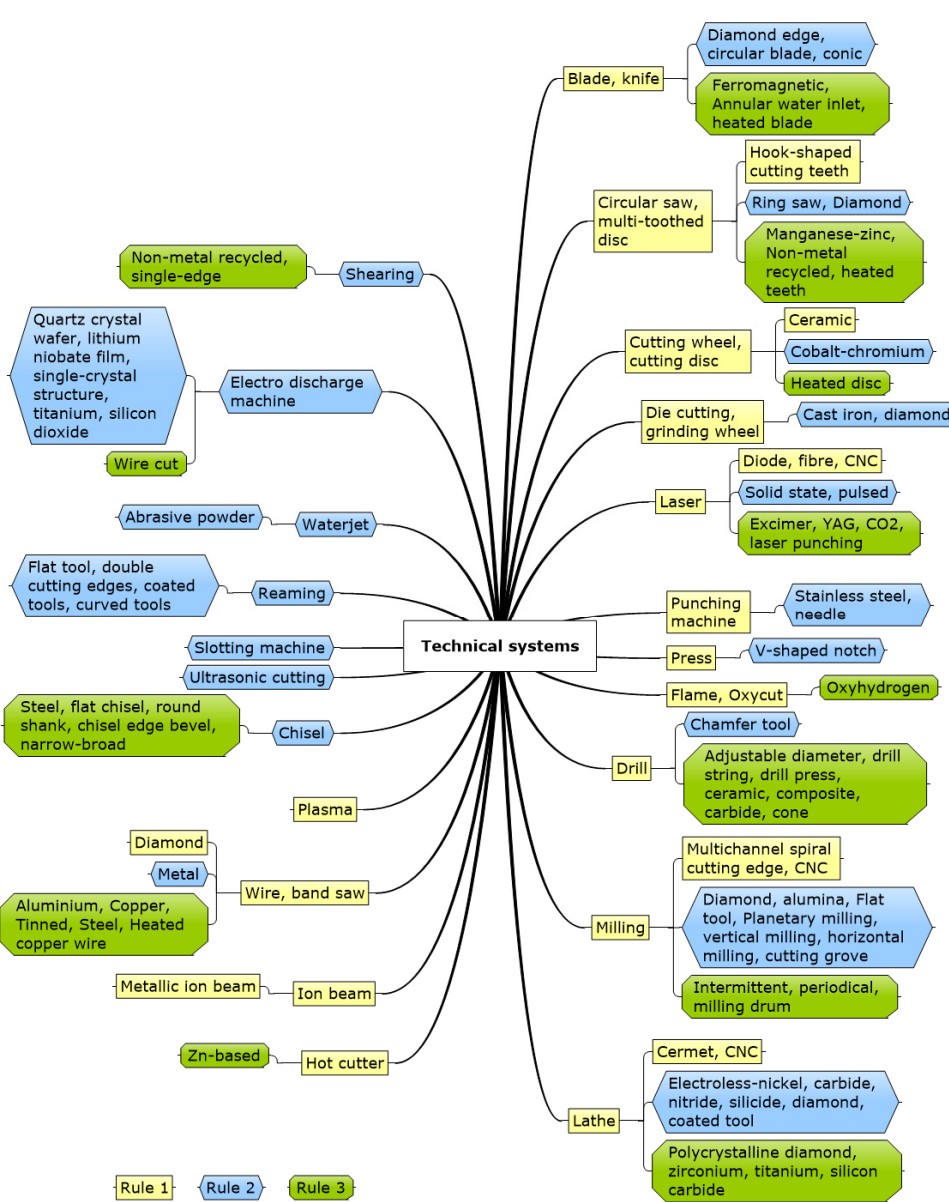

**Figure 4.** Technical system extracted by using the different proposed rules and divided into two levels of detail.

**Table 11.** Technical systems identified only with the rules using only syntactic analysis, classified according to their level.

| Level | Technical Systems |
| --- | --- |
| 1st level (16) | Blade, knife, circular saw, multi-toothed disc, cutting wheel, die cutting, laser, punching machine, press, flame, oxycut, milling, lathe, wire, plasma, waterjet. |
| 2nd level (9) | Ceramic cutting wheel, grinding wheel, laser CNC, spiral cutting edge, milling CNC, cutting groove, lather cermet, lathe CNC, diamond wire. |

**Table 12.** Missing Technical Systems.

| Missing Technical Systems | Related Sentences |
|---|---|
| 1. Conic multi-edge cutter | The conicity of the multi-edge cutter is 10–30 degrees (CN206215803). |
| 2. Rake cutting blade | The cutting-rake angle of the main cutting blade is 5–30 degrees. (CN206065448) |
| 3. Drill rod with circumferential distributed cutting edges | The cutting edges are distributed along the circumferential direction of the drill rod (CN205733152). |
| 4. Spiral diamond wire | Diamond wire is wound in the spiral grooves of the wire-winding drum (CN205969535). |

This choice was made because, as has been formulated, the proposed method without additional dependency patterns could be automatable as it is, while their use would introduce among the technical systems many elements that are not automatable, as explained in Section 3.2.2, and that would considerably lower the value of the method's precision. While the first test of precision is therefore a rigorous evaluation of the method, the second is instead an indication of the need to develop an automatism, currently missing, for the additional dependency patterns.

The first test showed that the precision of the method is 98%, since 101 technical systems were extracted, of which 99 were valid, out of the total 117 valid, presented in Section 4.2. The two technical systems that were discarded are reported in Table 13, together with the sentences from which they were extracted.

**Table 13.** Wrong technical systems automatically extracted.

| Wrong Technical Systems | Sentences |
|---|---|
| Coolant apparatus | A coolant apparatus for cutting metal (CN106994622) |
| Cylindrical metal workpiece | A cylindrical metal workpiece by the laser beam to cut the metal while holding and rotating the workpiece (JP6342442) |

Although the sample provided in Table 13 is extremely small with regard to drawing conclusions about the reasons for these errors, it is still possible to comment on them in order to understand them. The cause of both is the incorrect way of writing the sentences. The first, by incorrectly presenting "coolant apparatus" as the subject of the verb "cut" (i.e., function), expressed with the dependency pattern "for + Function+ing", and "metal" as the object (i.e., object), actually cheated the system in the application of Rule 1, which returned "coolant apparatus" among the technical systems. On the other hand, the convoluted form of the second sentence did not identify "laser beam" as a technical system, but "cylindrical metal workpiece". The reason in this case could be the automatic translation from Japanese, i.e., the language of the patent, to English, performed by the database (i.e., Orbit) from which the text of the document was exported.

The second test showed that by using the additional semantic patterns of Rule 2 and Rule 3, 47 technical systems were found, of which 18 were new, compared to those found with the method without the additional semantic patterns, and were therefore included in the final list (see Figure 4). These technical systems are reported in 1187 sentences, within which there are also 142 other structural components, but not technical systems, which, by simply automating the method, would be erroneously understood as such, lowering the overall precision of the method. The selection criterion in this case was purely manual.

*4.5. Second Case Study*

In this case study, the considered function is "process" while the object is "manure". The analysed pool still consists of 5000 selected patents, with the same criteria as in the previous case, based on the most recent priority date of those from the year 2016, starting from the query: "((digest+ or elimat+ or transform+ or process+) 2d (manure? or dung? or

muck? or dropping? or sewage?) and (eprd = 2016))/TX", in full text. The selected patents were analysed in the same way.

All the results obtained confirmed the goodness of the proposed method, even if in this case with slightly lower values than the previous case, albeit highly positive.

A total of 43 technical systems were identified. Of these, 20 were identified with Rule 1, to which 15 new ones from Rule 2 and 8 from Rule 3 were added. Analysing the level of detail of the technical systems, it emerged that 11 belonged to the first level, of which 9 were obtained with Rule 1 and 2 with Rule 2. Of the 32 from the second level, 11 derived from Rule 1, 13 from Rule 2 and 8 from Rule 3. Also, in this case, therefore, Rule 2 and Rule 3 proved to be fundamental for significantly increasing the results of Rule 1, and especially for exploring the second-level ones.

The analysis of the usefulness of the dependency patterns showed that 16 (i.e., 37%) technical systems were determined without them, or with only syntactic analysis, confirming also, in this case, the usefulness of this option.

Finally, the values associated with recall and precision of the method in this case study are both around 90%. The recall was obtained from the ratio of the technical systems identified to the method, and for those extracted manually it was equal to 90%, since five technical systems present in the selected pool were not identified by the method. Meanwhile, the precision was 91% considering only the technical systems automatically extracted, without considering those deriving from the additional dependency patterns in a semi-manual way. This precision value corresponds to the ratio of the technical systems obtained in this way and considered valid (31) to the total ones extracted in the same way (34).

## 5. Conclusions

This study proposed a method to automatically extract technical systems that perform a given function on a given object, from patents. The method is based on three novel rules, which in turn include three assumptions: sentence selection, syntactic analysis based on SAO extraction, and the use of dependency patterns. The main novelty concerns the use of multiple sentences, in which syntactic and semantic relations are sought, to increase the number of collected results. Although the method has been conceived to support design, it can also be used to support prior art searches and a literature background.

The advantages of the method, i.e., high precision and recall in the identification of technical systems, were demonstrated with two case studies about metal cutting and manure processing. In particular, the high recall was favoured by overcoming traditional syntactic analysis through dependency patterns and by analysing multiple sentences. In addition, the heterogeneity and the level of detail of the obtained results were increased by retrieving information about the many characteristics of technical systems, i.e., materials, structural parts, and their combinations.

Based on the obtained results, the method can be recommended to designers with different levels of experience and with different design needs. In traditional design, once the problem to be solved has been formulated according to a functional logic, as typically suggested by design theories, the proposed method can support the identification of the alternative solutions, while, if used iteratively, the method suggest new ideas to guide the innovative design process. Firstly, using generic functions and objects, the support provided by the method is mostly exploratory. Then, the analysis of the retrieved technical systems can highlight new problems to be solved, which can be reformulated with other functions and objects to use as new input in the method. Functional research is also useful for formulating many problems, such as those related to eco-design and the circular economy, as shown with the second case study.

The proposed method can have different repercussion concerning the improvement of the patent search for prior art, knowledge retrieval for design, and for data collection for the foreground inventory in prospective LCAs (e.g., [32]).

The main advantages of the proposed method are ease of use, since the proposed dependency patterns can be introduced directly into the query, and the categorization of the results. In particular, through the categorization, the designer can classify ready-to-use solutions on two levels of detail: structural and functional. This makes the subsequent problem-solving/design activity easier and faster.

However, the method still has some limitations. The ability to correctly formulate/reformulate the initial problem according to the functional logic from which to extract the query to build the pool of documents for analysis, is left to the user's experience. The combination between the proposed method and the existing tools for automatic text analysis produces, in fact, a semi-manual analysis, since the research between multiple sentences is not supported. In addition, the results obtained from some dependency patterns, i.e., those called auxiliaries, require a manual check. Such tasks can be time-consuming with current technologies, especially when many patents are analysed. The links of the multiple sentences in a patent based on present relationships can easily be automated in a short time with the established rules, although the proposed method does not implement them. On the other hand, the selection of technical systems from those obtained with auxiliary dependency patterns cannot be automated, and requires a certain knowledge of the application field. Another limitation is that this classification is often not made with academic and scientific criteria, but mainly with professional and labour-market criteria, so this creates distortions in the independent scientific approach. Finally, the proposed method has been tested only in case studies with calculable differences. For this reason, future developments should include applying the method in other case studies, in order to further refine it.

In addition, other possible future improvements can be reached through the integration of the method with artificial intelligence. The ever-greater reliability of artificial intelligence can in fact increasingly replace the semi-manual analysis of the content of the collected patents, to test their relevance.

**Author Contributions:** Conceptualization, C.S. and M.S.; methodology, C.S. and M.S.; validation, C.S. and M.S.; formal analysis, C.S. and M.S.; investigation, C.S. and M.S.; data curation, C.S. and M.S.; writing—original draft preparation, C.S. and M.S.; writing—review and editing, C.S. and M.S.; visualization, C.S. and M.S.; supervision, C.S. and M.S. All authors have read and agreed to the published version of the manuscript.

**Funding:** This study received no funding.

**Data Availability Statement:** All data are available on request.

**Conflicts of Interest:** The authors declare no conflict of interest.

## Appendix A

**Table A1.** Extended ontology of the Minimal Technical System model.

| Ontological Elements | Definitions |
| --- | --- |
| Technical system | The machine (e.g., a device or a plant) searched by the designer to solve the design problem. It is responsible for performing the function, or for transforming the object into the product. In turn, the technical system consists of different parts, which are classified at a lower level as supply, transmission, tool, and control, according to the minimal technical system model. The objective for which the technical system is considered in TRIZ is the paradoxical one of eliminating it, or at least eliminating its parts, ideally solving the problem of transforming the object into a product. To do this, a greater involvement of the resources present in the operating environment is sought during the problem solving/design activity. |
| Function | The action performed by the technical system. It may be sufficient (for the designer or the user), insufficient, missing, or harmful, as the case may be. These definitions depend on how the designer or user evaluates the object–product transformation, that is, the requirements of the resulting product. |
| Object | The entity on which the technical system performs the function. |

<p align="center">**Table A1.** *Cont.*</p>

| Ontological Elements | Definitions |
|---|---|
| Product | The transformed object after the function has been performed. The definition of the characteristics of the product, with respect to those of the object, i.e., the requirements of the problem, are the basis of the design activity, i.e., the design/modification/research of the technical system. |
| Tool | The part of the technical system that is in direct contact with the object during the performance of the function. The contact can be mechanical, acoustical, thermal, chemical, electrical, magnetic, intermolecular, or biological. This depends on the mode of transmission of energy from the tool to the object during the realization of the function. In turn, energy is supplied to the tool by supply through transmission, and generated by the supply, while controls regulate the flow of energy between the parts. They can do this in several ways, such as by varying the geometry of the tool. |

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
