# Peer review of "A Set of Rules for Function-Oriented Automatic Multi-Sentence Analysis in Patents"

_knowledge, doi:10.3390/knowledge3030025_

Round 1
Reviewer 1 Report
In this manuscript, the authors propose three rules for function-oriented automatic multi-sentences analysis in patents. This work may benefit some researchers via extracted patent information. Overall, I recommend it for publication after minor revisions:
1. The authors should carefully check the grammar. For example, in the second part of Literature review, the authors state “their ability to analyze only single sentences to extract relations among elements”. But “sentences” should be a singular form.
2. The abbreviation of Generic Technical system should be mentioned firstly in Table 6 instead of Table 8.
3. In Fig. 4, the authors list technical systems extracted by using the different proposed rules. As a researcher, I think that it is not enough to just extract the technical systems without functions. It is hard for me to understand efficient information just according to Fig. 4. Could you combine different methods to extract both technical systems and functions?
4. The authors mention some limitations related with the proposed method. Could you add some sentences to envision possible improvements?
/
Reviewer 2 Report
Dear Authors,
the world of patents is very complex and extracting information is not always simple. As indicated by the authors, a simplified approach is needed that can support designers in the search for information. The idea of determining a set of rules to extract information is very interesting and could bring advantages in the industrial world.
However, the following changes are recommended:
Is it possible to check if the rules are correct? With incorrect rules what kind of information do they get?
Can the rules be general and be applied to different sectors?
Can rules be implemented in an automatic search system?
What are the main advantages proposed by the method? it is necessary to highlight this aspect
Moderate editing of English language required
Reviewer 3 Report
The paper is well structured and sound. Literature is adequate and thoroughly investigated. Limitations of the research are explained.
Small details follow...
Why was the TRIZ model chosen, what were alternatives and what criteria was used for its selection?
Also, I find no need to reference Sections in the Introduction, as it was done in first paragraph. The last paragraph is sufficient for that.
The text needs proofreading. Some grammatical structures are strange (e. g. second sentence in the Introduction starts with "Their texts" - it is unclear who are *they*; "step consists in extracting" - of extracting maybe?, etc.).
Reviewer 4 Report
This work is well within the scope of Knowledge, and it may be of interest to most of the readers of this journal. The manuscript shows an introductory background material sufficient for someone not an expert in this area to understand the context and significance of this work, with good references to follow, especially in the field of IoT devices in daily life, capable of acquiring biomedical and biometric parameters.
It was clear that the main contribution of this work was to proposes a method that aims to establish some rules on how to perform a function-oriented search (providing function and object) to extract technical systems from patents, using syntax and dependency patterns for multi-sentence analysis.
The proposed method allows to identify relationships between technical system, function, and object even when these elements are disseminated in different sentences. Obviously, the objective of the method is to improve the quality and quantity of the results obtained in a fully automatic way, reducing the final control by the user.
That is very clear from the authors, and I totally agree, that this is a real novelty and the most effective target.
The main research hypothesis is to verify, with the case study, if the values of precision, i.e. the percentage of relevant results out of all those found, and recall, i.e. the percentage of results found out of all possible results, are acceptable. In addition, the usefulness, according to the stated objective, of integrating dependency patterns with respect to syntactic analysis and the use of multiple sentences compared to single ones is also investigated.
A weakness of this article is the fact that the authors in their attempt to verify, with only two case studies, if the values of precision, i.e. the percentage of relevant results out of all those found, and recall, i.e. the percentage of results found out of all possible results.
I also disagree with the attempt to rephrase the goal of the manuscript. As the authors write " The objective of this article can therefore be reformulated, in line with this ontology, as the automatic determination of Technical systems from patents, to perform a Function on Object assigned by the user.", this cannot be accepted, after the detailed reference and description in the abstract. I understand the author’s goal, but it needs to be worded differently, otherwise it creates misunderstandings.
According to my opinion and this researchers team, by manually analyzing all the documents it was possible to identify exactly all the Technical systems present in the analyzed pool, which was equal to 97%. At the second case the values associated with recall and precision of the method in this case study are both around 90%. That alone is a pretty big difference. You do not agree?
Also the technical objects that sometimes lead to the creation of patents are usually quite clear in its description. This is usually not the case in other scientific data, such as in the social or medical sciences and elsewhere. Therefore, especially at this point there is an additional great difficulty in generalizing them.
In conclusion, because the verification was done only in two cases and with calculable differences, it is mandatory to accept the results, but only as guiding searches and to generalize them as a method only if they are verified in various areas and in a much larger number of checks and searches. Especially this last conclusion, in my opinion, should be added as a limitation of this research.
Turnitin returned a similarity index of 14% including the references, while this dropped to 6% when I excluded it, so it’s marginally OK.
The English in this paper is good, except for some grammatical mistakes across the text that need proofreading.
In conclusion, this article and its scientific conclusions are very interesting.
Finally, for all the above and the specific comment below, I have opted to recommend the Acceptance of this manuscript after a Minor Revision.
Specific comments
Abstract
P2, L9: ‘The application on two case studies…’ -> please revise.
1. Introduction
P4, L15: ‘the proposed method allows to identify relationships between technical system,’ -> please revise.
2. Literature review
P4, L(at the end): ‘In fact, existing methods obtaining only parts of these relations, because they exploit only some of these dependencies’ -> please revise.
P5, L10: ‘However, their main limitations are the domain-specificity and the need to continuous updates.’ -> please revise.
Another limitation is that this classification is often not made with academic and scientific criteria, but mainly with professional and labor market criteria, so this creates distortions in the independent scientific approach.
3. Related Works
Consider the case that your proposal description on page 5, which is very useful for understanding the system, as a block diagram (Minimal Technical system model), be made even more detailed in an appendix.
3.1 Problem space and pool definition
P7, L1: ‘This because those available’ à please revise.
4.1 Test execution
P14, L32: ‘Sketch Engine’ à A reference should be provided for the statement.
5. Conclusions
P20, L13: ‘even if it requires the use of an additional tool.’à please explain.
The English in this paper is good, except for some grammatical mistakes across the text that need proofreading.
